# An effective and rapidly degradable disinfectant from disinfection byproducts

Jiarui Han [1,2], Wanxin Li [1,2] & Xiangru Zhang [1] ✉

Chloroxylenol is a worldwide commonly used disinfectant. The massive consumption and relatively high chemical stability of chloroxylenol have caused eco-toxicological threats in receiving waters. We noticed that chloroxylenol has a chemical structure similar to numerous halo-phenolic disinfection byproducts. Solar detoxification of some halo-phenolic disinfection byproducts intrigued us to select a rapidly degradable chloroxylenol alternative from them. In investigating antimicrobial activities of disinfection byproducts, we found that 2,6-dichlorobenzoquinone was 9.0–22 times more efficient than chloroxylenol in inactivating the tested bacteria, fungi and viruses. Also, the developmental toxicity of 2,6-dichlorobenzoquinone to marine polychaete embryos decreased rapidly due to its rapid degradation via hydrolysis in receiving seawater, even without sunlight. Our work shows that 2,6-dichlorobenzoquinone is a promising disinfectant that well addresses human biosecurity and environmental sustainability. More importantly, our work may enlighten scientists to exploit the slightly alkaline nature of seawater and develop other industrial products that can degrade rapidly via hydrolysis in seawater.

Since it was first reported in late December 2019, the coronavirus disease (COVID-19) has caused millions of confirmed deaths[1]. The COVID-19 pandemic has greatly impacted human life and changed human activities. One of the most discernible impacts has been the proliferation of personal and environmental disinfection measures to prevent the transmission of diseases[2–5]. According to a report by Reckitt Benckiser[6], its global sales of disinfectant products have increased by over 50% from a pre-pandemic value of US\$ 5.45 billion per year. Because chemical disinfectants may end up in natural aquatic environments via wastewater effluent and surface runoff, concerns have been raised that intensive disinfection could trigger secondary disasters in aquatic ecosystems[7]. Therefore, environment-friendly and effective disinfectants are urgently needed[8,9].

Chloroxylenol, also known as *para*-chloro-meta-xylenol (PCMX), is a halogen-substituted phenolic disinfectant. As with phenol, mechanisms of phenolic disinfectants to inactivate pathogens involve disruption of the cell membrane and impairment of microbial enzyme systems[10], while the halogenated derivatives of phenol tend to have

better antimicrobial activity than phenol[11]. Since its development in the 1920s, PCMX has been widely used as a broad-spectrum antimicrobial agent. The use of PCMX has been further increased in recent years[12]. PCMX has been found in 16.9% of antiseptic detergents in the United States[13], 20.7% of household cleaners in the United Kingdom[14], and 56.3% of household disinfectants and 33.9% of hand sanitizers in China[12]. Due to its extensive use and relatively high chemical stability, PCMX has been frequently detected in aquatic environments, e.g., at 0.1–1.2 µg L$^{-1}$ in Indonesian river water[15], 0.2–10.6 µg L$^{-1}$ in river water in Hong Kong[16], 1.62–9.57 µg L$^{-1}$ in river water in China (mainland)[12], and 0.06–0.79 µg L$^{-1}$ in seawater in Kuwait[17]. While PCMX is generally considered safe for humans[18], the U.S. Environmental Protection Agency (USEPA) has indicated that PCMX is moderately toxic to aquatic invertebrates and highly toxic to freshwater fish[19]. Toxicological studies have reported adverse effects of PCMX on aquatic organisms, including endocrine disruption, embryonic mortality, and malformations[20–22]. Chronic exposure to PCMX at environmental concentrations (~4.2 µg L$^{-1}$) can cause gene regulation and morphological

[1]Department of Civil and Environmental Engineering, The Hong Kong University of Science and Technology, Hong Kong SAR, China. [2]These authors contributed equally: Jiarui Han, Wanxin Li. ✉e-mail: xiangru@ust.hk

changes in rainbow trout[20]. Recently, we have detected and identified ~120 halogenated disinfection byproducts (DBPs) in disinfected drinking water and wastewater effluents[23]. We noticed that most of the newly identified DBPs are halo-phenolic compounds that are structurally similar to PCMX, such as halophenols, halonitrophenols, halohydroquinones, halohydroxybenzaldehydes, halohydroxybenzoic acids, and halosalicylic acids (Fig. 1). Other research teams have also identified some novel halo-phenolic DBPs, including halobenzenetriols and halohydroxybenzonitriles[24–27]. The structural properties of halo-phenolic DBPs appear to enable their antimicrobial activities as disinfectants. Additionally, our previous studies on the transformation of the newly identified DBPs[28,29] have shown that some halo-phenolic DBPs can be detoxified by solar photolysis through dehalogenation and further ring-cleavage to aliphatic compounds in receiving seawater. This implies that the use of halo-phenolic DBPs as disinfectants could mitigate the ecological problems caused by disinfectants. However, the efficacy of halo-phenolic DBPs in inactivating pathogens, the degradability of halo-phenolic DBPs in aquatic environments without sunlight, and the associated toxicity variation remain unknown.

Here, we investigated the efficacy of different DBPs against typical pathogenic microorganisms in comparison with PCMX. By coupling the disinfection efficiency with the degradation and detoxification kinetics in receiving seawater, a potential disinfectant was selected out of these DBPs. We discovered that the selected DBP exhibited substantially stronger antimicrobial efficacy than PCMX and that its concentration and associated developmental toxicity in receiving seawater decreased rapidly, even in darkness. Our results demonstrate the potential of this DBP as an effective and broad-spectrum disinfectant that is rapidly degradable and detoxified in receiving seawater.

## Results

### Antimicrobial activities of DBPs

Initially, we screened halo-phenolic DBPs for potential disinfectants primarily based on their acute toxicity and degradability[28–31]. Nitrogenous DBPs such as halonitrophenols and halohydroxybenzonitriles were excluded because they generally have higher toxicity and higher stability than carbonaceous DBPs[26,32]. Among the newly identified halo-phenolic DBPs, 5-bromosalicylic acid and 2,5-dibromohydroquinone exhibited the shortest half-lives under solar irradiation[29]. 2,4-Dihalophenols exhibited relatively high photodegradation rate constants and low developmental toxicity among halophenols[29,31]. Recent advances in DBP studies have indicated that iodinated DBPs are generally more toxic but less stable than their brominated and chlorinated analogs[33–35]. Therefore, the chlorinated, brominated, and iodinated species of the three groups of DBPs (i.e., 2,4-dihalophenols, 2,5-dihalohydroquinones, and 5-halosalicylic acids) were included (Fig. 1).

We first examined the antimicrobial activities of the selected DBPs and PCMX against *Escherichia coli* (gram-negative bacteria), which is the commonly used indicator of pathogenic contamination and has been identified as the driving agent for colorectal cancer[36]. By following the USEPA-recommended time range of 0.25–10 min for surface disinfection[37], a contact time of 5 min was selected. The disinfection efficiencies against *E. coli* were, in descending order, 2,4-dihalophenols > 2,5-dihalohydroquinones > 5-halosalicylic acids (pH 7.2, Fig. 2a). The survival curves of *E. coli* followed the delayed Chick-Watson law[38], with 2,4-diiodophenol (0.79 L mg$^{-1}$ h$^{-1}$) having a higher inactivation rate constant than PCMX (0.66 L mg$^{-1}$ h$^{-1}$), which in turn had a higher inactivation rate constant than other DBPs (Supplementary Table 1). Considering that disinfectant products are usually diluted with municipal water (pH 6.5–8.5) before use, the inactivation of *E. coli* by the DBPs was examined under different pH conditions. With increasing pH, the disinfection efficiencies of 2,4-dihalophenols remained the same, while those of 2,5-dihalohydroquinones increased significantly by 1.8–3.2 times (Fig. 2b). Since 5-halosalicylic acids were found to be ineffective in inactivating *E. coli* at the concentrations and pH values tested (Fig. 2a, b), they were excluded from further testing. To verify the broad-spectrum antimicrobial potential of the DBPs, we examined their performance in inactivating *Staphylococcus aureus* (gram-positive bacteria), *Candida albicans* (fungi), and bacteriophage MS2 (viruses) at pH 7.2 and contact time 5 min. The efficacy ranking order of PCMX and the DBPs in inactivating *S. aureus*, *C. albicans* or MS2 was basically the same as that in inactivating *E. coli*, with 2,4-diiodophenol showing the greatest antimicrobial effect at the same dose (Fig. 2c).

**Fig. 1 | Selection of DBPs for potential disinfectants.** Schematic illustration of screening halogenated phenolic DBPs for potential disinfectants based on their structural properties and photodegradation kinetics.

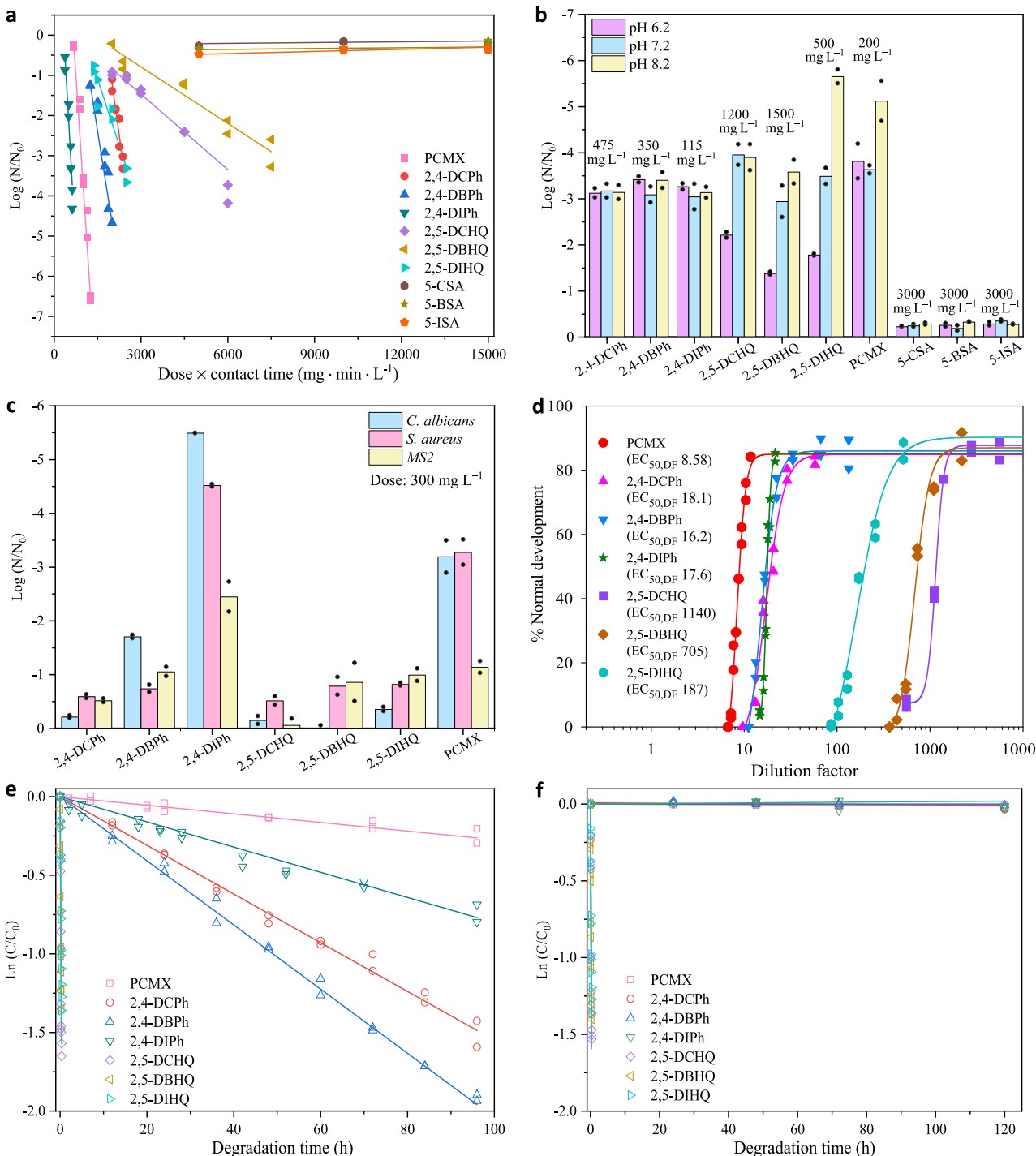

**Fig. 2 | Antimicrobial activities, developmental toxicity, and degradation of PCMX and nine DBPs. a** *E. coli* survivorship against the dosage of PCMX and the DBPs at pH 7.2 ($10^7$ cells per sample). **b** The effect of pH on disinfection efficacy of PCMX and the DBPs ($10^7$ *E. coli* cells per sample). **c** Inactivation of *C. albicans*, *S. aureus* and MS2 by PCMX and the DBPs (pH 7.2, contact time 5 min, dose 300 mg L$^{-1}$, $10^7$ cells per sample for *C. albicans* and *S. aureus*, $10^{10}$ plaque-forming units per sample for MS2). **d** Comparative developmental toxicity of PCMX and the DBPs to *P. dumerilii* embryos (150 embryos per sample). **e** Degradation of PCMX and the DBPs in seawater with solar irradiation. **f** Degradation of PCMX and the DBPs in seawater without solar irradiation. 2,4-Dichlorophenol (2,4-DCPh), 2,4-dibromophenol (2,4-DBPh), 2,4-diiodophenol (2,4-DIPh), 2,5-dichlorohydroquinone (2,5-DCHQ), 2,5-dibromohydroquinone (2,5-DBHQ), 2,5-diiodohydroquinone (2,5-DIHQ), 5-chlorosalicylic acid (5-CSA), 5-bromosalicylic acid (5-BSA), 5-iodosalicylic acid (5-ISA). The data presented were from $n = 2$ independent experiments. Source data are provided as a Source Data file.

## Toxicity of DBPs to marine polychaete embryos

Seawater is the immediate or ultimate receiving water body for municipal wastewater and urban runoff from coastal or inland areas. Urban wastewater has been wreaking havoc on marine ecosystems for decades[39,40]. *Platynereis dumerilii* is a ubiquitously distributed marine polychaete in coastal waters, where it feeds on marine algae and plays a vital role at the base of the trophic pyramid in marine ecosystems. It has been widely used as a model organism for marine ecotoxicity assessment[29,31]. To evaluate the risk of using DBPs as disinfectants for *P. dumerilii* embryonic development, we considered both the usage

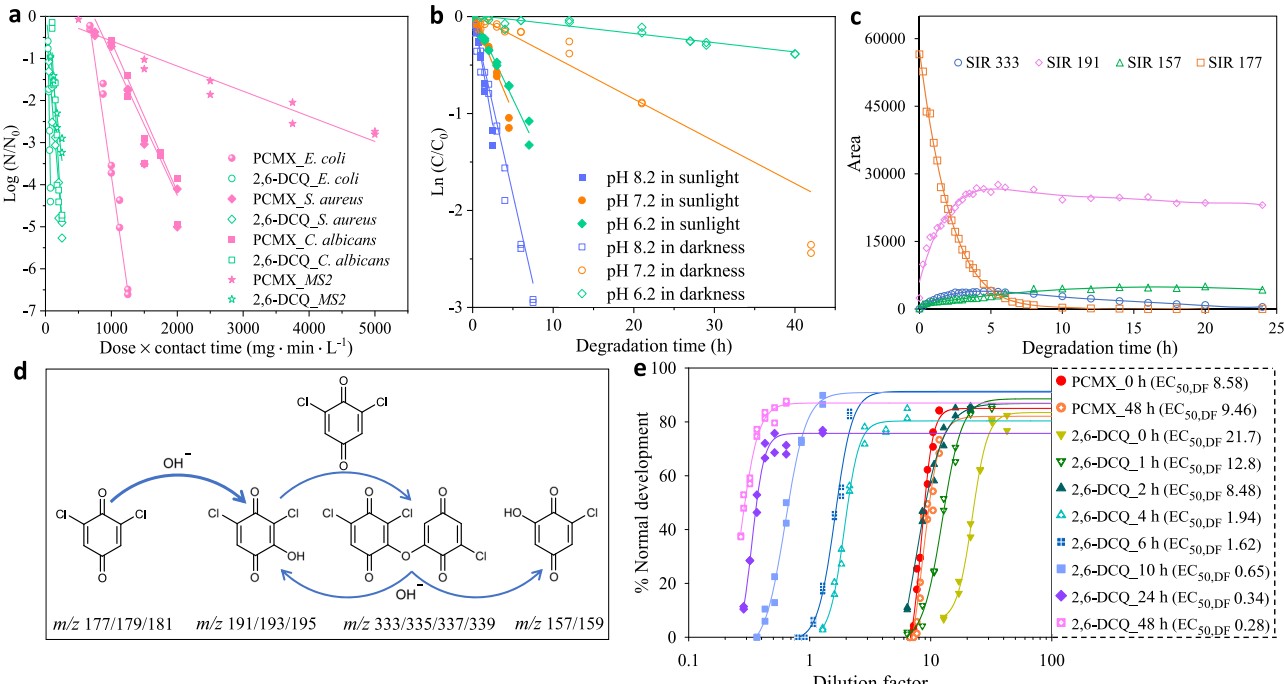

**Fig. 3 | Antimicrobial activity, degradation kinetics and pathway, and developmental toxicity of 2,6-DCQ. a** Survivorship of *E. coli*, *S. aureus*, *C. albicans*, and MS2 against the dosage of 2,6-DCQ and PCMX at pH 7.2 ($n = 2$ independent experiments, $10^7$ cells per sample for *E. coli*, *C. albicans* and *S. aureus*, $10^{10}$ plaque-forming units per sample for MS2). **b** Degradation of 2,6-DCQ under different pH conditions with and without solar irradiation ($n = 2$ independent experiments). **c** Peak area variations of 2,6-DCQ and its degradation products with time. **d** A proposed degradation pathway of 2,6-DCQ in seawater. **e** Comparative developmental toxicity of 2,6-DCQ and PCMX solutions with different degradation times in darkness ($n = 2$ independent experiments, 150 *P. dumerilii* embryos per sample). Source data are provided as a Source Data file.

scenario and the toxicity potency of PCMX and the DBPs. Taking the dose for 3-log (99.9%) reduction of *E. coli* (Supplementary Table 1) as the in-use concentration, we determined the dilution factor (DF) of this concentration at which 50% of the embryos developed normally compared with the control sample (EC$_{50,DF}$, Fig. 2d). A disinfectant with a higher EC$_{50,DF}$ requires a larger dilution factor to attenuate its developmental toxicity, indicating a higher risk. The EC$_{50,DF}$ was in descending order of 2,5-dihalohydroquinones > 2,4-dihalophenols > PCMX. In receiving seawater, PCMX and the DBPs were degraded following pseudo-first-order kinetics ($R^2 > 0.972$, Supplementary Table 2) under sunlight or in darkness. The solar photolysis of 2,4-dihalophenols (half-lives of 35–83 h, Fig. 2e) was faster than that of PCMX (half-life of 257 h). In the absence of sunlight, both 2,4-dihalophenols and PCMX were degraded slightly (less than 5% in 120 h, Fig. 2f). 2,4-Dihalophenols exhibited 1.9–2.1 times higher EC$_{50,DF}$ and 3.1–7.3 times greater solar decay constants than PCMX, suggesting that 2,4-dihalophenols could be potential disinfectants. Considering that the UV component of sunlight can penetrate only the uppermost meters of seawater[28] and degradation of 2,4-dihalophenols occurs mainly via solar photolysis in the surface fraction of seawater in the sunny daytime, 2,4-dihalophenols are still not ideal enough to be used as disinfectants. In comparison, 2,5-dihalohydroquinones were extremely rapidly degraded under light and dark conditions (Fig. 2e, f), with half-lives of 0.12–0.16 h. Because of the rapid degradation, the EC$_{50,DF}$ values of the chlorinated, brominated and iodinated 2,5-dihalohydroquinones were calculated to be lower than that of PCMX after 0.92, 0.94, and 0.71 h, respectively, in seawater in darkness. However, the extremely high instability in water (pH 6.2–8.2, Supplementary Fig. 1) and the high in-use concentrations in disinfection (Supplementary Table 1) rendered 2,5-dihalohydroquinones unsuitable as disinfectants. Inspiringly, we noted that with increasing pH, the disinfection efficiencies of 2,5-dihalohydroquinones increased drastically (Fig. 2b). Hydroquinone is susceptible to oxidation in alkaline solutions[41], which

led us to hypothesize that dihalobenzoquinones (oxidation products of dihalohydroquinones) might be more efficient in inactivating pathogens.

## Superior antimicrobial activities of 2,6-dichlorobenzoquinone (2,6-DCQ)

To test this hypothesis, we investigated the antimicrobial properties of 2,6-dichlorobenzoquinone (2,6-DCQ), which is also a newly identified DBP[42,43] and was found to be slightly more efficient in inactivating *E. coli* than its isomer 2,5-DCQ (Supplementary Table 3). 2,6-DCQ exhibited substantially higher disinfection efficacy against all four pathogens tested than PCMX (Fig. 3a). The inactivation rate constants of 2,6-DCQ and PCMX against *E. coli*, *S. aureus*, *C. albicans*, and MS2 were determined (Supplementary Table 4). At a fixed contact time of 5 min, doses of 12.8, 28.6, 35.4, and 46.1 mg L$^{-1}$, respectively, were required for 2,6-DCQ to achieve 3-log inactivation of *E. coli*, *S. aureus*, *C. albicans* and MS2 (Fig. 3a); the corresponding doses required for PCMX were 188, 308, 320 and 1010 mg L$^{-1}$, respectively, which were 9.0–22 times higher than the doses required for 2,6-DCQ. To gain insights into the disinfection mechanism of 2,6-DCQ, we stained *E. coli* with propidium iodide, a membrane-impermeable dye, and performed a fluorescence microscopy analysis. Disinfection with 2,6-DCQ resulted in a substantial number of propidium iodide-stained *E. coli* (Supplementary Fig. 2), indicating that *E. coli* cells lost membrane integrity after exposure to 2,6-DCQ. A higher fluorescence intensity indicates a greater number of cells with a damaged membrane. The increased fluorescence intensity (3.5–59.2 times that of the control sample) with increasing the 2,6-DCQ dose indicates that disruption of cell membrane plays an important role in disinfection with 2,6-DCQ.

## Rapid degradation and detoxification of 2,6-DCQ in seawater

To address the potential environmental concern regarding the use of the proposed disinfectant, we investigated the degradation of 2,6-DCQ

in receiving seawater (pH 8.2). It followed pseudo-first-order kinetics ($R^2 > 0.963$, Fig. 3b and Supplementary Table 5) with and without solar irradiation. Intriguingly, the degradation rate of 2,6-DCQ in darkness was comparable to that under sunlight. In seawater, the half-life of 2,6-DCQ under sunlight (1.33 h) was 193 times shorter than that of PCMX (257 h), and the half-life of 2,6-DCQ in darkness (1.74 h) was 3000 times shorter than that of PCMX (~240 days). To shed light on the rapid degradation of 2,6-DCQ in seawater, we then investigated the effect of pH on the degradation kinetics of 2,6-DCQ (Fig. 3b). By adjusting the seawater to pH 6.2 and 7.2, the half-lives of 2,6-DCQ under solar irradiation were 4.13 and 3.20 h, respectively; the corresponding half-lives of 2,6-DCQ in darkness were 76.2 and 13.4 h, respectively (Supplementary Table 5). The results indicate that sunlight is effective in promoting the degradation of 2,6-DCQ, with the effect being more pronounced at lower pH. The enhanced degradation under sunlight might be due to photonucleophilic substitution[28] and reactions with generated reactive oxygen species (e.g., hydroxyl radicals)[44]. Additionally, the degradation of 2,6-DCQ was dramatically accelerated with increasing pH, indicating the important role of hydrolysis in 2,6-DCQ degradation, particularly in darkness. We also investigated the effect of the high chloride level in seawater on the degradation of 2,6-DCQ in darkness. The results showed that chloride ions had no discernible effect on the degradation of 2,6-DCQ (Supplementary Fig. 3). The rapid degradation of 2,6-DCQ in seawater was attributed to the slightly alkaline nature (pH 8.2) of seawater, where the hydroxide concentration is 15.8 times higher than that at pH 7.0, and enhanced hydrolysis of 2,6-DCQ can occur. We further explored the degradation pathway of 2,6-DCQ in seawater in darkness. Using ultra-performance liquid chromatography/electrospray ionization-triple quadrupole mass spectrometry (UPLC/ESI-tqMS), we detected three degradation products of 2,6-DCQ in seawater in darkness (Supplementary Fig. 4). We also synthesized the most significant degradation product, 3-hydroxyl-2,6-dichloro-1,4-benzoquinone (OH-DCQ), and verified the purity of the synthesized compound with UPLC/ESI-tqMS (Supplementary Fig. 5). With the drastic degradation of 2,6-DCQ ($m/z$ 177/179/181, Fig. 3c), OH-DCQ ($m/z$ 191/193/195) rapidly formed via hydrolysis of 2,6-DCQ. The concentration of OH-DCQ reached its maximum at 5.5 h, corresponding to 41% of the initial molar concentration of 2,6-DCQ, and then remained stable (Supplementary Fig. 6). A molecular ion ($m/z$ 333/335/337) also formed rapidly, and it was proposed to be a dimeric product of OH-DCQ coupled to the parent compound (Fig. 3d). The coupling product could undergo hydrolysis to form 3-hydroxyl-6-monochloro-1,4-benzoquinone ($m/z$ 157/159) and OH-DCQ, and thus its peak area maximized with a relatively short degradation time (3.75 h). Hydrolysis of the coupling product was evidenced by the stable peak area of OH-DCQ after depletion of the parent compound.

Considering the doses required for 3-log $E.$ $coli$ inactivation, we evaluated the comparative risks of 2,6-DCQ and PCMX on $P.$ $dumerilii$ embryo development (Fig. 3e). At the point of discharge into seawater (with a degradation time set as 0 h), 2,6-DCQ showed slightly higher toxicity than PCMX. However, with the rapid degradation of 2,6-DCQ in seawater in darkness, the toxicity of 2,6-DCQ (essentially a mixture of 2,6-DCQ and its degradation products) decreased dramatically, with $EC_{50,DF}$ decreasing from 21.7 to 0.28 in 48 h. We also evaluated the developmental toxicity of OH-DCQ (Supplementary Fig. 7), the primary degradation product of 2,6-DCQ. The developmental toxicity of 2,6-DCQ to $P.$ $dumerilii$ embryos was 149 times higher than that of OH-DCQ, indicating that the degradation of 2,6-DCQ in seawater is a rapid detoxification process. By contrast, the toxicity of PCMX remained unchanged from 0 to 48 h, which was consistent with the negligible degradation of PCMX in darkness. Specifically, after 2 h of degradation, 2,6-DCQ and PCMX were comparable in toxicity, with $EC_{50,DF}$ of 8.48 and 8.58, respectively. As the degradation time increased, the $EC_{50,DF}$ of 2,6-DCQ continued to decrease and reached only 3.0% of the $EC_{50,DF}$ of PCMX after 48 h; in

other words, 2,6-DCQ was 31 times less toxic than PCMX after 48-h degradation in darkness.

## Discussion

Disinfection practices ensure biosecurity for humans but pose challenges for environmental sustainability. This dilemma has been underscored during the global pandemic COVID-19, which has necessitated intensive practices of personal and environmental disinfection. PCMX is a disinfectant widely used in antiseptic detergents, household disinfectants and hand sanitizers before, during, and after the global pandemic. The widespread use of this disinfectant and its relatively high chemical stability have led to ecological threats in receiving water bodies. We noticed that PCMX has a similar chemical structure to numerous halo-phenolic DBPs that we have previously identified. Our previous finding[28] on the detoxification of some DBPs with sunlight exposure intrigued us to select a rapidly degradable disinfectant from halo-phenolic DBPs. First, we selected three groups of DBPs (2,4-dihalophenols, 2,5-dihalohydroquinones, and 5-halosalicylic acids) from >100 newly identified halo-phenolic DBPs based on their degradability and toxic potency. We examined the disinfection performance of the selected DBPs against pathogens and their (photo) degradation kinetics in receiving seawater. The extremely high instability of 2,5-dihalohydroquiones in water and the ineffectiveness of 5-halosalicylic acids in inactivating pathogens made them unsuitable as disinfectants. 2,4-Dihalophenols exhibited comparable disinfection power to PCMX and could be potential disinfectants. However, the relatively slow degradation of 2,4-dihalophenols in the absence of solar irradiation and the related potential ecological risk prevented them from being considered ideal disinfectants. Second, inspired by the unexpected variation of the decay rates and disinfection efficiencies of 2,5-dihalohydroquinones with pH, we hypothesized that dihalobenzoquinones (i.e., the oxidation products of dihalohydroquinones) might be powerful antimicrobial agents. Using 2,6-DCQ as a representative, we found that its disinfection efficiency was 9.0–22 times higher than that of PCMX in inactivating typical indicator microorganisms, indicating that 2,6-DCQ is an effective disinfectant. Third, we demonstrated the high degradability of 2,6-DCQ in water, particularly in alkaline aquatic environments. Degradability of toxic substances is critical for the sustainability of ecosystems[45]. In seawater, 2,6-DCQ was readily degraded with a half-life of 1.74 h, even without solar irradiation. The enhanced hydrolysis and detoxification of 2,6-DCQ in the marine environment was ascribed to the slightly alkaline environment of seawater (~pH 8.2). Forty-eight hours after discharge into seawater, 2,6-DCQ exhibited 31 times lower toxicity to $P.$ $dumerilii$ embryos than PCMX. Finally, we disclosed the degradation pathway of 2,6-DCQ in seawater without solar irradiation and synthesized the primary degradation product of 2,6-DCQ, OH-DCQ, which showed extremely lower developmental toxicity than the parent compound. The literature[28,29,31] indicates that photoconversion of OH-DCQ can lead to further substitutions of hydroxyl groups on the benzene ring and eventual cleavage of the benzene ring to form aliphatic compounds; more importantly, each step in the conversion of OH-DCQ to aliphatic compounds is a detoxification one.

It is noteworthy that despite its rapid degradation in alkaline solutions, 2,6-DCQ can remain stable at room temperature for over 7 months in 2-propanol (a common component in antiseptic liquids and hand sanitizers) and for over 36 months in solid form (Supplementary Fig. 8), which suits it for storage, transportation, and wide application. 2,6-DCQ can be used directly in a 2-propanol solution. It can also be prepared in the form of concentrated 2-propanol solutions or solid pills or tablets, which will be diluted or dissolved in water before use. We also conducted a comparative cost analysis for 2,6-DCQ and PCMX. Based on the consumption of raw materials and energy in the synthesis process[46,47], the cost for producing 2,6-DCQ should be lower than that for producing PCMX (Supplementary Fig. 9). In

general, 2,6-DCQ may be used as an alternative in the occasions where PCMX is used as an antiseptic or a disinfectant, including but not limited to personal care products, such as hand cleansers, detergent, and soap; other products such as paint, textile, metal working fluids, and medical scrubs; sanitation of surgical instruments and food equipment; sanitation in public places like hospitals, clinics, shopping centers, restaurants, and streets; and sanitation in the household. It needs mentioning that although 2,6-DCQ exhibited stronger antimicrobial activities than PCMX, according to the Hodge and Sterner toxicity scale (including extremely toxic, highly toxic, moderately toxic, and slightly toxic)[48], both chemicals are classified as substances "slightly toxic" to humans based on their median lethal doses to mammalian animals (LD$_{50}$; oral, rat)[49,50]. In addition, the much stronger antimicrobial activities of 2,6-DCQ than PCMX indicate much lower in-use concentrations of 2,6-DCQ when it is used as a disinfectant. Considering the comparable toxic potency of the two chemicals and the lower in-use concentrations of 2,6-DCQ, the toxicity potential of 2,6-DCQ to humans is probably lower than that of PCMX. Even so, the safety precautions that apply to PCMX or other disinfectants should also apply to 2,6-DCQ. Future studies may be conducted to systematically evaluate the human toxicity and environmental impact of 2,6-DCQ.

Our study presents a promising solution to the dilemma of disinfection in the context of human biosecurity pressures and eco-sustainability challenges. The results indicate the applicability of 2,6-DCQ as a promising alternative to the commonly used PCMX for disinfection. The strong disinfection efficacy of 2,6-DCQ and its rapid detoxification in seawater may allow it to impact the multi-billion-dollar global disinfectant market and to better support the biosecurity of human society and the sustainability of the aquatic environment. More importantly, this study disclosed that the rapid degradation and detoxification of 2,6-DCQ in seawater in the absence of sunlight is due to enhanced hydrolysis in seawater, which is slightly alkaline (pH 8.2) and has a stable hydroxide concentration that is about 16 times higher than at neutral pH. This seems to be part of the beauty of nature. By taking advantage of the slightly alkaline environment of seawater, scientists may design and develop other industrial products such as pesticides, pharmaceuticals and personal care products that can be rapidly degraded by hydrolysis in receiving seawater. These may contribute significantly to achieving two of the Sustainable Development Goals established by the United Nations, namely to "ensure access to safe water, sanitation and hygiene" (the most basic human need for health and well-being) and to "conserve and sustainably use the oceans, seas and marine resources" (the provision of marine and coastal biodiversity for the livelihoods of over three billion people)[51].

## Methods

### Antimicrobial activities of DBPs

The chemicals and reagents used are detailed in Supplementary Note 1. The bacteria *E. coli* (ATCC 25922) and *S. aureus* (ATCC 25923) were cultured aerobically in Luria-Bertani broth (Sigma-Aldrich) and Tryptone Soya broth (Oxoid), respectively, at 37 °C for 16 h. The fungus *C. albicans* (ATCC 14053) was aerobically cultured in Sabouraud Dextrose broth (Difco) at 37 °C for 24 h. The microorganism cells were harvested by centrifugation at 1200×g (10 min, 4 °C) and resuspended in phosphate-buffered saline (PBS, 10 mM, pH 7.2). The disinfection experiments were performed in PBS solutions (10 mM, pH 7.2) with an initial concentration of *E. coli*, *S. aureus*, or *C. albicans* of 10$^6$–10$^7$ cells per milliliter. A contact time of 5 min was used in accordance with the recommended time range of 0.25–10 min by USEPA for surface disinfection[37]. Preliminary tests were conducted to determine the appropriate range of DBP doses. The appropriate dose range was established when the lowest and highest doses could achieve <1-log and >3-log of microbial inactivation, respectively. Based on the preliminary results, a series of doses of DBPs were added to the cell

suspensions at room temperature (22 ± 1 °C). After the contact time of 5 min, the microorganism cells were enumerated following a membrane filter method[52]. Briefly, serial 10-fold dilutions were conducted for the disinfected samples and the control sample without disinfection. Each diluted suspension was subjected to filtration through 0.45-μm membrane, which was then placed on a petri dish. The Petri dishes containing *E. coli*, *S. aureus*, and *C. albicans* were incubated in m-FC medium (Becton Dickinson & Co) at 45 °C for 24 h, in Tryptone Soya broth at 37 °C for 24 h and in Sabouraud Dextrose broth at 37 °C for 24 h, respectively. After incubation, the colony-forming units within a statistically reasonable range (20–80) were counted.

The bacteriophage MS2 (ATCC 15597-B1) was propagated in the host *E. coli* (ATCC 15597) in Tryptone Soya broth at 37 °C for 16 h. The viruses were separated from the bacterial debris and purified by centrifuging the phage culture (4000×g, 10 min, 4 °C) and filtering the lysate with a 0.2 μm polyethersulfone sterile filter. The filtrate was then diluted in PBS (10 mM, pH 7.2) to an initial MS2 concentration of 10$^9$–10$^{10}$ plaque-forming units per millimeter. To initiate the disinfection experiments, a series of DBP doses were added to the MS2 suspension at room temperature (22 ± 1 °C). After a contact time of 5 min, the viruses were immediately diluted 10$^3$ times in Tryptone Soya broth (the 1000-fold diluted DBP dose was shown to insignificantly affect MS2 and the host *E. coli* for a contact time of 16 h) and enumerated following a double-layer agar method[53]. Briefly, serial 10-fold dilutions were conducted for the disinfected samples and the control sample without disinfection. Each dilution was mixed with the host *E. coli* in the soft agar and the mixture was immediately spread onto a Petri dish with Tryptone Soya agar. After the Petri dishes were incubated at 37 °C for 16 h, the plaque-forming units were counted. The disinfection experiments were performed in duplicate. All the materials for the disinfection experiments were autoclaved before use.

The impact of pH on the disinfection efficiencies of DBPs was examined. *E. coli* (ATCC 25922) was used as the test strain. The disinfection experiments were performed in 10 mM PBS solutions (pH 6.2, 7.2, and 8.2) with an initial *E. coli* concentration of 10$^6$–10$^7$ cells per milliliter. To initiate the disinfection, a DBP was dosed with a bacteria suspension at a predetermined concentration. After the given contact time (5 min) at room temperature (22 ± 1 °C), the bacteria were enumerated following the membrane filter method[52]. The bacteria concentrations in the 10 mM PBS solutions (pH 6.2, 7.2, and 8.2) without the addition of DBPs were used as controls.

### Degradation of DBPs in seawater

The solar photolysis experiments were conducted in a chamber equipped with eight simulated sunlight full-spectrum lamps (BlueMax 5900, Full Spectrum Solutions). The temperature in the chamber was maintained at 20 °C. The total intensity of simulated sunlight was determined using a digital photometer (DT-1010B, Shenzhen Golden Octopus, China). Quartz flasks were positioned in the chamber where the total intensity of simulated sunlight was 4.79 ± 0.04 mW cm$^{-2}$ for each quartz flask. Such light intensity was comparable to 10% of the average solar intensity on seawater surface in sunny daytime in Hong Kong[28]. A DBP was added at 10 mg L$^{-1}$ to 50 mL pretreated seawater (pH 8.2) in the quartz flasks and exposed to the simulated solar irradiation for 0–96 h. Every 8 h, the quartz flasks were rotated in and out of the chamber, and the sample volume was determined and adjusted with ultrapure water to account for any water evaporation. During the solar irradiation experiments, the sample was adjusted to pH 6.2 and 7.2 or maintained at pH 8.2 by adding 4 mM phosphate buffer (pH 6.2, 7.2, and 8.2, respectively).

To study the DBP degradation in seawater in darkness, a DBP was added at 10 mg L$^{-1}$ to 50 mL pretreated seawater (pH 8.2) in the quartz flasks. The flasks were wrapped with aluminum foil to simulate the dark condition. The samples were kept at 20 °C in darkness for 0–120 h, during which the sample was adjusted to pH 6.2 and 7.2 or maintained

at pH 8.2 by adding 4 mM phosphate buffer (pH 6.2, 7.2, and 8.2, respectively).

For the DBP samples with and without sunlight, aliquots were taken at a specific reaction time. The sample aliquots were analyzed using Agilent high-performance liquid chromatography (HPLC) equipped with a Poroshell 120 EC-C18 column (4.6 × 150 mm, 4 μm particle size, Agilent) and a UV detector. The mobile phase, consisting of water (pre-adjusted to pH 2 with phosphoric acid) and acetonitrile at 50/50 ($v/v$), was applied at a flow rate of 1.0 mL min$^{-1}$. PCMX, 2,4-dihalophenols and 5-halosalicylic acids were detected at the wavelength of 230 nm. 2,5-Dihalohydroquinones and 2,6-DCQ were detected at the wavelength of 280 nm. All the degradation experiments were conducted in duplicate.

## Developmental toxicity assay with the embryos of *P. dumerilii*

The DBP doses for 3-log *E. coli* reduction (pH 7.2) were set as the in-use concentrations, which were serially diluted and added to the *P. dumerilii* embryos at 12-h postfertilization. A dilution factor was defined as the ratio of the in-use concentration to the exposure concentration. Each exposure set contained a control sample (i.e., the embryos at 12-h postfertilization developed in seawater without the addition of DBPs). By 24-h postfertilization, normally developed embryos should have reached the first larval stage. An inverted stereomicroscope (×40 magnification) was used to observe the larvae. The normal development percentage was calculated as the ratio of normally developed embryos to total embryos. By plotting the curve of the normal development percentage versus the dilution factor, the EC$_{50,DF}$ value (i.e., the dilution factor corresponding to 50% of the normal development percentage of the control sample) was obtained using SigmaPlot 12. The developmental toxicity test was conducted in duplicate.

## UPLC/ESI-tqMS analysis

A Waters UPLC system coupled with ESI-tqMS was used to characterize the degradation products of 2,6-DCQ. The UPLC separation was performed using an HSS T3 column (100 × 2.1 mm, 1.8 μm particle size, Waters) with a column temperature of 35 °C. A gradient eluent of methanol–water was applied at a flow rate of 0.4 mL min$^{-1}$. The gradient program was set as follows: methanol/water ($v/v$) ratio changed linearly from 5/95 to 90/10 in the first 8 min and then returned to 5/95 during 8.0–8.1 min, which was held for 2.9 min for re-equilibration. The ESI-tqMS was set as follows: ESI negative mode, capillary voltage 2.8 kV, cone voltage 10–50 V (depending on analytes), cone gas flow 50 L h$^{-1}$, source temperature 120 °C, desolvation temperature 400 °C, desolvation gas flow 800 L h$^{-1}$, collision energy 15–50 eV (depending on analytes), collision gas (argon) flow 0.25 mL min$^{-1}$, and mass resolution 15 (1-unit resolution).

The degradation of 2,6-DCQ in seawater was simulated by adding 10 mg L$^{-1}$ 2,6-DCQ to 50 mL ultrapure water, which was adjusted to pH 8.2 with 5 mM sodium carbonate (to simulate the alkalinity in seawater and also to be compatible with the subsequent UPLC/ESI-tqMS analysis). The sample was kept in darkness for 0–24 h, during which the pH of the sample was maintained at 8.1 ± 0.1. At a designated reaction time, 7.5 μL of the sample was injected into the UPLC system. For a nonhalogenated degradation product, the selected ion recording (SIR) mode with a dwell time of 0.05 s was adopted to obtain the retention time. For halogenated degradation products, the multiple reaction monitoring mode with a dwell time of 0.05 s was used to obtain the retention time and the isotopic abundance ratio. To investigate the time-dependent profile of 2,6-DCQ and its degradation products, the peak area of each compound was determined using the SIR mode in the UPLC/ESI-tqMS analysis.

## Fluorescence microscopic analysis of *E. coli*

The *E. coli* cells in the control and disinfected samples were pelleted by centrifugation at 8000×$g$ (5 min, 4 °C) and resuspended in 200 μL PBS (0.1 M, pH 7.2). Propidium iodide[54] was added to the cell suspension at a concentration of 75 μM. After propidium iodide staining in darkness for 30 min, the samples were washed twice with PBS and mounted on glass slides. The samples were then examined using a fluorescence microscope (Eclipse Ti, Nikon, Japan). The fluorescence intensity was semi-quantified with ImageJ software (National Institutes of Health, Bethesda, MD).

## Synthesis of OH-DCQ

OH-DCQ was synthesized in our laboratory because the standard compound was not commercially available. Briefly, 2,6-DCQ (300 mg L$^{-1}$ in ultrapure water) was allowed to hydrolyze at room temperature. After 72 h, the sample was subjected to HPLC separation (Poroshell 120 EC-C18 column, 4.6 × 150 mm, 4 μm particle size, Agilent). The mobile phase consisting of water (pH 2, adjusted with hydrochloric acid) and methanol (60/40, $v/v$) was applied at a flow rate of 1.0 mL min$^{-1}$. Hydrochloric acid was used for water acidification to preclude the impact on the toxicity test with *P. dumerilii* embryos[55]. Three dominant peaks were detected at a wavelength of 280 nm (Supplementary Fig. 5a). The peaks at retention times of 4.94 and 6.12 min were confirmed with standard compounds as 2,6-dichlorohydroquinone and 2,6-DCQ, respectively. The peak at 2.69 min was proposed to be OH-DCQ, and the corresponding fraction (2.60–3.00 min) was collected for each injection. The sample injection and fraction collection were repeated 100 times. The collected solutions were divided into three aliquots. Each aliquot was concentrated and finally dried to a solid by rotary evaporation (180 rpm, 30 °C).

The first aliquot was subjected to UPLC/ESI-tqMS analysis to test the identity and purity of the collected fraction. For this purpose, the aliquot was dissolved in 2 mL ultrapure water, acidified to pH 2 with sulfuric acid (10%, $v/v$), added with sodium sulfate to saturation, and extracted with 2 mL methyl tert-butyl ether (MtBE). The MtBE layer was rotoevaporated to 0.5 mL and then added with 20 mL acetonitrile. The mixture was rotoevaporated to 0.5 mL and stored at 4 °C. Prior to UPLC/ESI-tqMS analysis, 0.5 mL of the aliquot in acetonitrile was mixed with 0.5 mL of ultrapure water. Only one peak was detected in the UPLC/ESI-tqMS full scan chromatogram of the collected fraction (Supplementary Fig. 5b). The mass spectrum at the retention time of 3.53 min showed an ion cluster $m/z$ 191/193/195 (Supplementary Fig. 5c), corresponding to OH-DCQ. The results confirmed the purity of the collected OH-DCQ.

The second aliquot was subjected to analysis of total organic halogen to quantify the collected OH-DCQ. The determination of total organic halogen was based on Standard Method 5320B[52], with modifications of using off-line ion chromatography for halide detection[56]. Briefly, the aliquot was dissolved in 100 mL ultrapure water and acidified to pH 2 with concentrated nitric acid[52] prior to adsorption onto pre-packed activated carbon columns (Mitsubishi). A three-channel adsorption module (TXA03C, Mitsubishi Chemical Analytech) was used for adsorption. An AQF-100 automatic quick furnace (Mitsubishi Chemical Analytech) was used for pyrolysis. During the pyrolysis process, the chlorine atoms in OH-DCQ were converted to chloride. The chloride ion concentration was determined using an ICS-3000 ion chromatography system (Dionex). 2,6-DCQ was used as a standard compound for measuring total organic halogen to determine the recovery of the analysis. The total organic halogen analysis was conducted in duplicate.

The third aliquot was subjected to the developmental toxicity assay with *P. dumerilii* embryos. The aliquot was dissolved in 10 mL of seawater and adjusted to pH 8.2. The embryos at 12-h postfertilization were exposed to different levels of OH-DCQ for 12 h. By 24-h postfertilization, normally and abnormally developed embryos were counted using an inverted stereomicroscope. Based on the percentage of normally developed embryos, an exposure−response curve was generated. The median effective concentration (EC$_{50}$) was determined

using SigmaPlot 12 software. The developmental toxicity of 2,6-DCQ was also evaluated for comparison. The toxicity bioassay was conducted in duplicate.

### Real-time stability testing of 2,6-DCQ

According to the U.S. Food and Drug Administration guidance[57], 2,6-DCQ in solid form or solution form (using 2-propanol as the solvent, a common ingredient in antiseptic liquids and hand sanitizers) was kept at room temperature ($22 \pm 1\,°C$) and $65 \pm 5\%$ of relative humidity for stability testing. The concentrations of 2,6-DCQ at different storage times were determined by an Agilent HPLC equipped with a UV detector at 280 nm. A disinfectant product is considered stable for a certain storage time if its concentration remains $\geq 90\%$ of the initial concentration[57].

### Reporting summary

Further information on research design is available in the Nature Portfolio Reporting Summary linked to this article.

## Data availability

The data supporting the findings of this work are included in the Supplementary Information/Source Data file. Source data are provided with this paper.

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

## Acknowledgements

This work was financially supported by the Research Grants Council of Hong Kong to X.Z. (No. 16212518).

## Author contributions

X.Z. conceived and supervised the project. J.H., W.L., and X.Z. designed the experiment. J.H. and W.L. performed experiments and analyzed data. J.H. and W.L. wrote the original draft. X.Z., J.H., and W.L. reviewed and edited the manuscript.

## Competing interests

The Hong Kong University of Science and Technology has filed patent applications (US Patent Application No. 18/522,414, pending; China Invention Patent Application No. 202311619379.X, pending) on this work, in which X.Z., J.H., and W.L. are co-inventors.
