## [Peer Review File · Nature Communications]

An effective and rapidly degradable disinfectant from disinfection byproductsREVIEWER COMMENTS

Reviewer #1 (Remarks to the Author):

The authors investigated the antimicrobial activities, developmental toxicities, and (environmental) degradability (or persistency) of some halobenzoquinones, in search for alternative greener disinfectant. The study began from noting that chloroxyenol (PCMX), a widely used antimicrobial agent, is structural analogues of halo phenolic disinfection byproducts (DBPs). The environmental degradability of halo-phenolic DBPs in aquatic systems (e.g, sunlit seawater) have previously been investigated (by the authors), showing that some phenolic DBPs undergo rapid degradation/detoxification. From this, a novel idea of using structural analogues of halo phenolic DBPs as environmental benign disinfectant was proposed and tested. Through systematic tests on antimicrobial activity, developmental toxicity, and environmental degradability with halo phenolic candidates, 2,6-dichlorobenzoquinone (2,6-DCQ) was found to be a promising green disinfectant, as 2,6-DCP exhibited higher antimicrobial activity (than PCMX), while underwent more rapid degradation in seawater, indicating low environmental persistency. It was also shown that 2,6-DCP undergoes facile hydrolysis at slightly basic pH (e.g., pH 8.2 as the pH of seawater) generating hydroxyl-dichlorobenzoquinone as major product with much lower toxicity.

Overall, the study is highly innovative, and demonstrates successful example of developing green disinfectant, considering environmental fate/impact. It is assumed that many of chemical ingredient used in current disinfectants are developed in old days, when environmental impact/sustainability was not the major consideration. With increasing demand on human biosecurity and eco-sustainability challenges, the green chemistry design approach should be more actively applied. In addition, the impact/significant of the study's finding can be huge, considering the size of global disinfectant market. Based on the novelty and significance of the findings, its publication in nature communication is highly recommended.

The reviewer has only some minor comments as below, which can be considered for the revision.

2,6-DCQ was shown to have higher degradability (meaning low stability) driven by hydrolysis compared to e.g, PCMX, a benchmark disinfectant. As a commercial product, stability of chemical product might also be important. Have the authors tested the stability of 2,6-DCQ in the condition/scenario of its application as commercial disinfectant?

Cost and overall environmental impact (through life cycle assessment) on mass production of 2,6-DCP and its storage/transport might also be carefully considered.

Are human toxicity (adverse effects) of halobenzoquinones well known (in comparison to PCMX)?

In Fig 1 and Fig 2, 2,4-dihalo or 2,5-dihalo compounds were mainly screened/tested, but in Fig 3, 2,6-DCP was finally selected. What was the main difference between 2,4 or 2,5 substitution vs 2,6 substitution?

Fig 3. Was hydroxyl-2,6-dichlorobenzoquinone quantified using standard? Otherwise, how do the authors know it as dominant product? Further degradation of this product would generate some haloaldehydes which might exhibit some (toxicological) concern?

Reviewer #2 (Remarks to the Author):

This is an interesting work and very well written manuscript that systematically tested the different disinfection byproducts for selecting a green disinfectant. They started with 3 classes of DBPs and compared their results with commercially used disinfectant PCMX. They systematically dropped the unsuitable DBPs and narrow down their study. Finally, they have shortlisted the 2,6 DCQ as a green disinfectant. They checked its anti bacterial activity against *E. coli*, *S. aureus* and *C. albicans*. They also evaluated the degradation time of these molecules to determine how long they will persist in water and their effect towards the aquatic life. These findings are important because

with increasing DBPs in water, we need disinfectant which is more environment friendly. However, there are some comments which authors need to address.

1. Authors have used different dosages of selected DBPs to study the anti-bacterial activity against E. coli. How was the dosage amount was selected for each DBP? How will the Fig. 2a,b look if the dosage of each DBP is same. Will the trend be same or different?
2. Page 2- 'Introduction' title is missing.
3. Authors have used different reagents to maintain the pH (such as for maintaining pH 8, somewhere they have used phosphate and then, sodium carbonate; similarly for pH 2, they have used HCl, H₂SO₄ and HNO₃). Is there any reason for that?
4. Authors have used sea water for studies. Is there any effect of different concentration of chloride ions on the activity or stability of 2,6 DCQ?
5. Earlier articles on toxicity of 2,6 DCQ have reported that even few µg/L concentration is toxic for human as well as aquatic lives (Environmental Pollution 245 (2019)719-724, Ecotoxicology and Environmental Safety 263 (2023) 115357). However; the authors have used up to 35 mg/L. How is its activity at lower concentrations?
6. Is 2,6 DCQ cost effective in comparison to commercial PCMX?

REVIEWER COMMENTS

Reviewer #1 (Remarks to the Author):

The authors investigated the antimicrobial activities, developmental toxicities, and (environmental) degradability (or persistency) of some halobenzoquinones, in search for alternative greener disinfectant. The study began from noting that chloroxyenol (PCMX), a widely used antimicrobial agent, is structural analogues of halo phenolic disinfection byproducts (DBPs). The environmental degradability of halo-phenolic DBPs in aquatic systems (e.g, sunlit seawater) have previously been investigated (by the authors), showing that some phenolic DBPs undergo rapid degradation/detoxification. From this, a novel idea of using structural analogues of halo phenolic DBPs as environmental benign disinfectant was proposed and tested. Through systematic tests on antimicrobial activity, developmental toxicity, and environmental degradability with halo phenolic candidates, 2,6-dichlorobenzoquinone (2,6-DCQ) was found to be a promising green disinfectant, as 2,6-DCQ exhibited higher antimicrobial activity (than PCMX), while underwent more rapid degradation in seawater, indicating low environmental persistency. It was also shown that 2,6-DCQ undergoes facile hydrolysis at slightly basic pH (e.g., pH 8.2 as the pH of seawater) generating hydroxyl-dichlorobenzoquinone as major product with much lower toxicity.

Overall, the study is highly innovative, and demonstrates successful example of developing green disinfectant, considering environmental fate/impact. It is assumed that many of chemical ingredient used in current disinfectants are developed in old days, when environmental impact/sustainability was not the major consideration. With increasing demand on human biosecurity and eco-sustainability challenges, the green chemistry design approach should be more actively applied. In addition, the impact/significant of the study's finding can be huge, considering the size of global disinfectant market. Based on the novelty and significance of the findings, its publication in nature communication is highly recommended.

The reviewer has only some minor comments as below, which can be considered for the revision.

RESPONSE: We are very grateful for your careful reading and appreciation of the manuscript and for your constructive suggestions.

2,6-DCQ was shown to have higher degradability (meaning low stability) driven by hydrolysis compared to e.g, PCMX, a benchmark disinfectant. As a commercial product, stability of chemical product might also be important. Have the authors tested the stability of 2,6-DCQ in the condition/scenario of its application as commercial disinfectant?

RESPONSE: A nice point. It is true that the stability of disinfectant products is important. We have assessed the stability of 2,6-DCQ using a real time testing method. In brief, we stored 2,6-DCQ in solid form (the 2,6-DCQ standard in solid form was purchased in 2020, 2022, and 2023,

respectively) or in solution form (with 2-propanol as solvent; stock solutions were prepared in December 2022 to December 2023) at room temperature with $65 \pm 5\%$ of relative humidity. We measured the concentration of 2,6-DCQ in the stored samples with different storage times. The results showed that for 2,6-DCQ in solid form, the concentration of 2,6-DCQ varied in $<2\%$ within 36 months; for the 2,6-DCQ solution in 2-propanol, the concentration of 2,6-DCQ remained $>90\%$ of the initial concentration for over seven months.

Regarding the application, we suggest packaging 2,6-DCQ in solid form or in solution form using solvents such as 2-propanol (a common component in antiseptic liquids and hand sanitizers). Thus, 2,6-DCQ can be used directly in a 2-propanol solution; it can also be prepared in the form of concentrated 2-propanol solutions or solid pills or tablets, which will be diluted or dissolved in water before use. This is just like the way of using PCMX products; for example, the Walch Multipurpose Disinfectant contains 4.5–5.5% PCMX, and the manufacturer suggests diluting the product to 1:150 with water for hand washing or to 1:100 with water for household hygiene.

To address your point, we have provided the results of the stability tests in the Discussion and provided the data in the Supplementary Materials.

In the Methods (lines 472–478 in the revision-marked manuscript),

“Real-time stability testing of 2,6-DCQ

According to the U.S. Food and Drug Administration guidance⁵⁷, 2,6-DCQ in solid form or solution form (using 2-propanol as the solvent, a common ingredient in antiseptic liquids and hand sanitizers) was kept at room temperature (22 ± 1 °C) and $65 \pm 5\%$ of relative humidity for stability testing. The concentrations of 2,6-DCQ at different storage times were determined by an Agilent HPLC equipped with a UV detector at 280 nm. A disinfectant product is considered stable for a certain storage time if its concentration remains $\geq 90\%$ of the initial concentration⁵⁷.”

Lines 264–270 in the revision-marked manuscript, “It is noteworthy that despite its rapid degradation in alkaline solutions, 2,6-DCQ can remain stable at room temperature for over seven months in 2-propanol (a common component in antiseptic liquids and hand sanitizers) and for over 36 months in solid form (Supplementary Fig. 8), which suits it for storage, transportation, and wide application. 2,6-DCQ can be used directly in a 2-propanol solution. It can also be prepared in the form of concentrated 2-propanol solutions or solid pills or tablets, which will be diluted or dissolved in water before use.”

Supplementary Fig. 8 | The concentrations of 2,6-DCQ in solution form (using 2-propanol as the solvent) and solid form at different storage times.

“57. U.S. Food and Drug Administration. Q1A(R2) Stability Testing of New Drug Substances and Products. FDA-2002-D-0222 (2003).”

Cost and overall environmental impact (through life cycle assessment) on mass production of 2,6-DCQ and its storage/transport might also be carefully considered.

RESPONSE: Thank you for the suggestion. We agree that cost is an important aspect that needs to be considered. Since 2,6-DCQ has not yet been mass produced for commercial application, we propose to evaluate the cost of 2,6-DCQ (in comparison with PCMX) by considering the materials and energy required in the synthesis processes^{46,47}. In brief, to obtain 1 kg of PCMX, 2.63 kg of 3,5-dimethylphenol is placed in a suitable vessel with a stirrer, and 1.31 kg of sulfuryl chloride (SO₂Cl₂) is slowly added. The temperature is kept at 40 °C during the reaction. When all the SO₂Cl₂ has been added, the reaction mixture is heated to 80 °C and the volatile impurities in it are removed by air blowing. Part of the PCMX is separated from the mother liquor during cooling, and further PCMX can be isolated by vacuum distillation of the mother liquor and further crystallization. A total of 1 kg of PCMX can be obtained. To obtain 1 kg of 2,6-DCQ, 1.18 kg of 2,4,6-trichlorophenol is dissolved in 3.59 kg of methanol. The dissolved 2,4,6-trichlorophenol is added dropwise to 1200 mL of 5% (w/w) nitric acid in a closed reaction vessel with a stirrer. The reaction is carried out at 20 °C with an oxygen supply of 2.0 atm (i.e., 0.013 kg O₂ for a 5 L reaction vessel). After a reaction time of 2 hours, 1 kg of 2,6-DCQ can be obtained by filtration of the mother liquor. The unit prices of the raw materials are obtained from a global supplier Alfa Aesar (note that unit prices can be significantly lower for larger quantities). According to the synthesis flowcharts shown below, the raw material cost for synthesizing 2,6-DCQ is significantly lower than that for synthesizing PCMX. Since no heating or cooling processes are required for synthesizing 2,6-DCQ, the energy consumption should also be lower than that for synthesizing PCMX. We have supplemented the cost analysis in the revised manuscript.

Lines 270–273 in the revision-marked manuscript, “We also conducted comparative cost analysis for 2,6-DCQ and PCMX. Based on the consumption of raw materials and energy in the synthesis processes^{46,47}, the cost for producing 2,6-DCQ should be lower than that for producing PCMX (Supplementary Fig. 9).”

Supplementary Fig. 9 | Synthesis flowcharts of (a) PCMX and (b) 2,6-DCQ. The unit prices of raw materials were obtained from a global supplier Alfa Aesar (<https://alfaesar.com/>).

- “46. Gladden, G. W. Preparation of 2-chlor-meta-5-xyleneol. US Patent No. 2,350,677 (1944).
47. Xu, Y. Preparation method of 1,4-cyclohexanedione. CN Patent No. 109942388A (2019).”

Regarding the environmental impact assessment of 2,6-DCQ through a comparative life cycle assessment of PCMX and 2,6-DCQ, the environmental inputs and outputs in the processes of production (e.g., the raw materials and energy consumption), transportation (transportation of raw material and final products), and release (e.g., the impacts on human and ecosystem from the release of the chemicals) should be encompassed. While the energy and fuel consumption in the production and transportation of 2,6-DCQ and PCMX can be estimated based on the synthesis processes as described above, the human toxicity and environmental impact for 2,6-DCQ have not been systematically evaluated. We have added some information in the revised manuscript to address potential concerns.

As to the human toxicity of 2,6-DCQ (lines 278–287 in the revision-marked manuscript), “It needs mentioning that although 2,6-DCQ exhibited stronger antimicrobial activities against pathogens than PCMX, according to the Hodge and Sterner toxicity scale (including extremely toxic, highly toxic, moderately toxic, and slightly toxic)⁴⁸, both chemicals are classified as substances “slightly toxic” to humans based on their median lethal doses to mammalian animals (LD₅₀; oral, rat)^{49,50}. In addition, the much stronger antimicrobial activities of 2,6-DCQ than PCMX indicate much lower in-use concentrations of 2,6-DCQ when it is used as a disinfectant. Considering the comparable toxic potency of the two chemicals and the lower in-use concentrations of 2,6-DCQ, the toxicity potential of 2,6-DCQ to humans is probably lower than that of PCMX. Even so, the safety precautions that apply to PCMX or other disinfectants should also apply to 2,6-DCQ.”

As to the ecotoxicity of 2,6-DCQ (lines 251–263 in the revision-marked manuscript), “Third, we demonstrated the high degradability of 2,6-DCQ in water, particularly in alkaline aquatic environment. Degradability of toxic substances in water is critical for the sustainability of ecosystems⁴⁵. In seawater, 2,6-DCQ was readily degraded with a half-life of 1.74 hours even without solar irradiation. The enhanced hydrolysis and detoxification of 2,6-DCQ in the marine environment was ascribed to the slightly alkaline environment of seawater (~pH 8.2). Forty-eight hours after discharge into seawater, 2,6-DCQ exhibited 31 times lower toxicity to *P. dumerilii* embryos than PCMX. Finally, we disclosed the degradation pathway of 2,6-DCQ in seawater without solar irradiation and synthesized the primary degradation product of 2,6-DCQ, OH-DCQ, which showed extremely lower developmental toxicity than the parent compound. The literature^{28,29,31} indicates that photoconversion of OH-DCQ can lead to further substitutions of hydroxyl groups on the benzene ring and eventual cleavage of the benzene ring to form aliphatic compounds; more importantly, each step in the conversion of OH-DCQ to aliphatic compounds is a detoxification one.”

We have also indicated the need for further studies on the human toxicity and environmental impact of 2,6-DCQ (lines 287–289 in the revision-marked manuscript): “Future studies may be conducted to systematically evaluate the human toxicity and environmental impact of 2,6-DCQ.”

- “28. Liu, J., Zhang, X. & Li, Y. Photoconversion of chlorinated saline wastewater DBPs in receiving seawater is overall a detoxification process. *Environ. Sci. Technol.* **51**, 58–57 (2017).
29. Liu, J. et al. Phototransformation of halophenolic disinfection byproducts in receiving seawater: kinetics, products, and toxicity. *Water Res.* **150**, 68–76 (2019).
31. Yang, M. & Zhang, X. Comparative developmental toxicity of new aromatic halogenated DBPs in a chlorinated saline sewage effluent to the marine polychaete *Platynereis dumerilii*. *Environ. Sci. Technol.* **47**, 10868–10876 (2013).
48. Hodge, H. C. & Sterner, J. H. Tabulation of toxicity classes. *Am. Ind. Hyg. Assoc.* **10**, 93–96 (1949).
49. National Research Council. The Chemical-Biological Coordination Center of the National Research Council (The National Academies Press, 1953).
50. Liebert, M. A. Final report on the safety assessment of chloroxylenol. *J. Am. Coll. Toxicol.* **4**, 147–169 (1985).”

Are human toxicity (adverse effects) of halobenzoquinones well known (in comparison to PCMX)?

RESPONSE: Thank you for the comment. We have also recognized the importance of understanding the potential human toxicity of 2,6-DCQ in comparison with PCMX. The human toxicity potential of a chemical depends on both the inherent toxic potency of the chemical and the dose of human exposure to the chemical. Mammalian animal toxicity tests are an important method for understanding and comparing the toxic potency of different chemicals to humans. The “Hodge and Sterner toxicity scale” is one of the most common scales for classifying chemicals. It classifies chemicals based on the oral median lethal doses (LD₅₀) to rats. According to the Hodge and Sterner toxicity scale⁴⁸, there are four categories of chemicals: extremely toxic (LD₅₀ < 1 mg/kg), highly toxic (LD₅₀ 1 to <50 mg/kg), moderately toxic (LD₅₀ 50 to <500 mg/kg), and slightly toxic (LD₅₀

500 to <5000 mg/kg). The lethal dose 10% (LD₁₀; the dose at which 10% of the individuals die) of 2,6-DCQ was 500 mg/kg (oral, rat)⁴⁹, indicating that the LD₅₀ of 2,6-DCQ was ~1000 mg/kg (oral, rat) or greater. The LD₅₀ of PCMX was reported to be 3830 mg/kg (oral, rat)⁵⁰. Therefore, both 2,6-DCQ and PCMX are classified as “slightly toxic” to humans. In addition, as we found that 2,6-DCQ was 9–22 times more efficient than PCMX in inactivating pathogens, the in-use concentration of 2,6-DCQ can be 9–22 times lower than that of PCMX. This indicates that human exposure to 2,6-DCQ would be much lower than that to PCMX when 2,6-DCQ is used as a disinfectant alternative to PCMX. Considering the comparable toxic potency of the two chemicals and the lower in-use concentrations of 2,6-DCQ, the toxicity potential of 2,6-DCQ to humans is probably lower than that of PCMX.

It should be noted that disinfectants are a class of chemicals with antimicrobial properties and should not be considered to be completely benign. The Hygiene Hub has recommended some precautions applicable to commonly used disinfectants (<https://resources.hygienehub.info/en/articles/4028561>), including: (1) Keep all chemical disinfectants in correctly labelled containers; (2) Do not mix chemical disinfectants together or with other cleaning products; (3) Avoid splashes and spills by handling chemical disinfectants with care; (4) Do not breathe vapor/gas or spray; prepare and use chemical disinfectants in well ventilated areas; (5) Only use water at room temperature for dilutions (unless specified otherwise in users’ instructions); (6) Do not eat, drink, or smoke when using chemical disinfectants; (7) Wear personal protective equipment (protective clothing, gloves and goggles) if available when handling chemical disinfectants; (8) Store chemical disinfectants out of reach of children and in a cool and dry place, protected from heat and sunlight; (9) Wash hands thoroughly with soap and water after handling chemical disinfectants; and (10) Do not use environmental/surface chemical disinfectants for personal hygiene. All these precautions should also be applicable to 2,6-DCQ.

To echo your point, we have indicated these in the revised manuscript:

Lines 278–287 in the revision-marked manuscript, “It needs mentioning that although 2,6-DCQ exhibited stronger antimicrobial activities against pathogens than PCMX, according to the Hodge and Sterner toxicity scale (including extremely toxic, highly toxic, moderately toxic, and slightly toxic)⁴⁸, both chemicals are classified as substances “slightly toxic” to humans based on their median lethal doses to mammalian animals (LD₅₀; oral, rat)^{49,50}. In addition, the much stronger antimicrobial activities of 2,6-DCQ than PCMX indicate much lower in-use concentrations of 2,6-DCQ when it is used as a disinfectant. Considering the comparable toxic potency of the two chemicals and the lower in-use concentrations of 2,6-DCQ, the toxicity potential of 2,6-DCQ to humans is probably lower than that of PCMX. Even so, the safety precautions that apply to PCMX or other disinfectants should also apply to 2,6-DCQ.”

“48. Hodge, H. C. & Sterner, J. H. Tabulation of toxicity classes. *Am. Ind. Hyg. Assoc.* **10**, 93–96 (1949).

49. National Research Council. *The Chemical-Biological Coordination Center of the National Research Council* (The National Academies Press, 1953).

50. Liebert, M. A. Final report on the safety assessment of chloroxylenol. *J. Am. Coll. Toxicol.* **4**, 147–169 (1985).”

In Fig 1 and Fig 2, 2,4-dihalo or 2,5-dihalo compounds were mainly screened/tested, but in Fig 3, 2,6-DCQ was finally selected. What was the main difference between 2,4 or 2,5 substitution vs 2,6 substitution?

RESPONSE: Thank you for the comment. Initially, we screened halo-phenolic DBPs for potential disinfectants primarily based on their acute toxicity and degradability (Fig. 1). For dihalophenols, we selected 2,4-dihalophenols instead of 2,6-dihalophenols because 2,4-dihalophenols showed larger photodegradation rate constants than 2,6-dihalophenols²⁹. 2,5-Dihalohydroquinones were selected because they exhibited the shortest half-lives under solar irradiation among tested halo-phenolic DBPs²⁹. Then, we were inspired by the results of degradation kinetics and antimicrobial tests of 2,5-dihalohydroquinones under different pH conditions (Fig. 2b and Supplementary Fig. 1), which led us to hypothesize that dihalobenzoquinones (oxidation products of dihalohydroquinones) might be more efficient in inactivating pathogens. In testing the hypothesis and conducting the subsequent experiments, we focused on 2,6-DCQ instead of 2,5-DCQ for two reasons. First, 2,6-DCQ has been newly identified as a DBP^{42,43}. Since 2,5-DCQ and 2,6-DCQ are isomers, the frequent detection of 2,6-DCQ as a DBP suggests that the yield of 2,6-DCQ during the synthesis process could be much higher than that of 2,5-DCQ. Second, we conducted preliminary tests on the antimicrobial activities of 2,5-DCQ and 2,6-DCQ, and our results showed that 2,6-DCQ was slightly more efficient in inactivating *E. coli* than 2,5-DCQ. We have supplemented the results in the Supplementary Materials and made revisions in the main text:

Lines 148–150 in the revision-marked manuscript, “To test this hypothesis, we investigated the antimicrobial properties of 2,6-dichlorobenzoquinone (2,6-DCQ), which is also a newly identified DBP^{42,43} and was found to be slightly more efficient in inactivating *E. coli* than its isomer 2,5-DCQ (Supplementary Table 3).”

Supplementary Table 3 | *E. coli* inactivation by 2,5-DCQ and 2,6-DCQ (pH 7.2, contact time 5 min).

Compound	Dose (mg L ⁻¹)	Log reduction (Mean ± SD)
2,5-DCQ	15	3.78 ± 0.40
	7.5	0.77 ± 0.14
2,6-DCQ	15	4.24 ± 0.22
	7.5	0.76 ± 0.16

“29. Liu, J. et al. Phototransformation of halophenolic disinfection byproducts in receiving seawater: kinetics, products, and toxicity. *Water Res.* **150**, 68–76 (2019).

42. Wang, W. et al. Analytical characterization, occurrence, transformation, and removal of the emerging disinfection byproducts halobenzoquinones in water. *TrAC Trends Anal. Chem.* **85**, 97–110 (2016).

43. Zhao, Y. et al. Occurrence and formation of chloro- and bromo-benzoquinones during drinking water disinfection. *Water Res.* **46**, 4351–4360 (2012).”

Fig 3. Was hydroxyl-2,6-dichlorobenzoquinone quantified using standard? Otherwise, how do the authors know it as dominant product? Further degradation of this product would generate some haloaldehydes which might exhibit some (toxicological) concern?

RESPONSE: Thank you for the comment. Since the standard compound of 3-hydroxyl-2,6-dichloro-1,4-benzoquinone (OH-DCQ) was not commercially available, we synthesized OH-DCQ in our laboratory. The details of synthesis, purity verification, and quantification of OH-DCQ were provided in the Methods (“Synthesis of 3-hydroxyl-2,6-dichloro-1,4-benzoquinone (OH-DCQ)”). In brief, a 300 mg L⁻¹ of 2,6-DCQ solution that had been kept at room temperature for 72 h was subjected to HPLC separation, and the eluent fraction corresponding to OH-DCQ (Supplementary Fig. 5a) was collected. The sample injection and fraction collection were conducted for 100 times. To test the identity and purity of the collected fraction, one aliquot of the collected fraction was extracted and then subjected to UPLC/ESI-tqMS analysis. The UPLC/ESI-tqMS confirmed the purity of the collected OH-DCQ (Supplementary Fig. 5b,c). To quantify the collected OH-DCQ, one aliquot of the collected fraction was subjected to the analysis of total organic chlorine. With the synthesized OH-DCQ standard and the commercial 2,6-DCQ standard, we could determine their concentrations during the degradation of 2,6-DCQ. The concentration of OH-DCQ first increased with 2,6-DCQ degradation and maximized at 5.5 hours, corresponding to 41% of the initial molar concentration of 2,6-DCQ. Regarding the effect of further degradation of OH-DCQ on the toxicity, we would like to clarify that we evaluated the toxicity of 2,6-DCQ solutions after degradation of 0–48 hours. The solutions are essentially a mixture of 2,6-DCQ and its degradation products, including the degradation products of OH-DCQ. The results showed that the toxicity of the 2,6-DCQ solution kept decreasing in the degradation process, and the toxicity after 48-hour degradation was 78 times lower than that at 0 hour. In addition, the literature^{28,29,31} indicates that photoconversion of OH-DCQ can lead to further substitutions of hydroxyl groups on the benzene ring and eventual cleavage of the benzene ring to form aliphatic compounds; more importantly, each step in the conversion of OH-DCQ to aliphatic compounds is a detoxification one.

To echo your comment, we have added the information on OH-DCQ in the revised manuscript. In addition, we have revised Fig. 3d to make the presented information more precise.

Lines 194–200 in the revision-marked manuscript, “We also synthesized the most significant degradation product, 3-hydroxyl-2,6-dichloro-1,4-benzoquinone (OH-DCQ), and verified the purity of the synthesized compound with UPLC/ESI-tqMS (Supplementary Fig. 5). With the drastic degradation of 2,6-DCQ (*m/z* 177/179/181, Fig. 3c), OH-DCQ (*m/z* 191/193/195) rapidly formed via hydrolysis of 2,6-DCQ. The concentration of OH-DCQ reached its maximum at 5.5 hours, corresponding to 41% of the initial molar concentration of 2,6-DCQ, and then remained stable (Supplementary Fig. 6).”

Lines 260–263 in the revision-marked manuscript, “The literature^{28,29,31} indicates that photoconversion of OH-DCQ can lead to further substitutions of hydroxyl groups on the benzene ring and eventual cleavage of the benzene ring to form aliphatic compounds; more importantly, each step of the conversion of OH-DCQ to aliphatic compounds is a detoxification one.”

Supplementary Fig. 6 | Concentrations of 2,6-DCQ and OH-DCQ with degradation time.

Fig. 3. d A proposed degradation pathway of 2,6-DCQ in seawater.

- “28. Liu, J., Zhang, X. & Li, Y. Photoconversion of chlorinated saline wastewater DBPs in receiving seawater is overall a detoxification process. *Environ. Sci. Technol.* **51**, 58–57 (2017).
29. Liu, J. et al. Phototransformation of halophenolic disinfection byproducts in receiving seawater: kinetics, products, and toxicity. *Water Res.* **150**, 68–76 (2019).
31. Yang, M. & Zhang, X. Comparative developmental toxicity of new aromatic halogenated DBPs in a chlorinated saline sewage effluent to the marine polychaete *Platynereis dumerilii*. *Environ. Sci. Technol.* **47**, 10868–10876 (2013).”

Reviewer #2 (Remarks to the Author):

This is an interesting work and very well written manuscript that systematically tested the different disinfection byproducts for selecting a green disinfectant. They started with 3 classes of DBPs and compared their results with commercially used disinfectant PCMX. They systematically dropped the unsuitable DBPs and narrow down their study. Finally, they have shortlisted the 2,6 DCQ as a green disinfectant. They checked its antibacterial activity against *E. coli*, *S. aureus* and *C. albicans*. They also evaluated the degradation time of these molecules to determine how long they will persist in water and their effect towards the aquatic life. These findings are important because with increasing DBPs in water, we need disinfectant which is more environment friendly.

However, there are some comments which authors need to address.

RESPONSE: We are very grateful for your careful reading and appreciation of the manuscript and for your constructive suggestions.

1. Authors have used different dosages of selected DBPs to study the anti-bacterial activity against *E. coli*. How was the dosage amount was selected for each DBP? How will the Fig. 2a,b look if the dosage of each DBP is same. Will the trend be same or different?

RESPONSE: Thank you for the comment. First, to evaluate the antimicrobial activity of DBPs against *E. coli*, we initially conducted preliminary tests to determine the appropriate dose range for each DBP. The lowest dose of an appropriate dose range could achieve <1-log of *E. coli* inactivation, and the highest dose of an appropriate dose range could achieve >3-log of *E. coli* inactivation. In the first batch of preliminary tests, the doses of PCMX and the selected DBPs were set at 0, 125, 250 and 500 mg L⁻¹; the results were used as reference values to increase or decrease the doses of PCMX and the DBPs in the subsequent batches of preliminary tests. Once the appropriate range was determined, a series of doses of PCMX and DBPs were added to *E. coli* cell suspensions at room temperature (22 ± 1 °C) for a contact time of 5 min to evaluate their disinfection efficiency. Second, we plotted the inactivation of *E. coli* at different dosages of PCMX and DBPs in Fig. 2a, from which the inactivation rate constants for PCMX and DBPs can be determined. The inactivation rate constants can be used to quantitatively compare the antimicrobial activity. According to the Chick-Watson law, when the dosage is the same, *E. coli* inactivation is proportional to the inactivation rate constants. Therefore, at the same dosage, the trend of disinfection efficiencies of PCMX and the DBPs is the same as the trend of their inactivation rate constants. Fig. 2b displays the impact of pH on the disinfection efficiencies of the tested chemicals for *E. coli*. As the disinfection efficiencies of PCMX and the DBPs were remarkably different (e.g., PCMX vs 2,5-dihalohydroquiones), the variation in disinfection efficiencies with pH would not be noticeable if the same dosage were used.

To echo your comment, we have supplemented the information on preliminary tests in the Methods of the revised manuscript:

Lines 321–326 in the revision-marked manuscript, “Preliminary tests were conducted to determine the appropriate range of DBP doses. The appropriate dose range was established when the lowest and highest doses could achieve <1-log and >3-log of *E. coli* inactivation, respectively. Based on the preliminary results, a series of doses of DBPs were added to the cell suspensions at room temperature (22 ± 1 °C).”

2. Page 2- ‘Introduction’ title is missing.

RESPONSE: Thank you for pointing this out. We have added the “Introduction” title in the revised manuscript.

3. Authors have used different reagents to maintain the pH (such as for maintaining pH 8, somewhere they have used phosphate and then, sodium carbonate; similarly for pH 2, they have used HCl, H₂SO₄ and HNO₃). Is there any reason for that?

RESPONSE: Thank you for the comment. When selecting chemicals to adjust or maintain pH, we generally followed common practices or established protocols, with the consideration of not affecting the reactions or subsequent analyses. Specifically, to maintain a near-neutral pH, both phosphate buffer and carbonate buffer are commonly used. For the degradation experiments, phosphate buffer was used to maintain pH 8 because in the subsequent HPLC analysis, the mobile phase consisted of water that was preset to pH 2; if carbonate buffer were used, carbon dioxide would be generated once the sample was mixed with the acidic mobile phase, which would affect the column pressure and thus the HPLC analysis. For the exploration of the degradation pathway, carbonate buffer was used to maintain pH 8 because in the subsequent UPLC/ESI-tqMS analysis, if phosphate buffer were used, the non-volatile phosphate would deposit in the sample cone in the MS system and thus damage the instrument. For the synthesis of OH-DCQ, as one aliquot of the collected eluent fraction would be subjected to the toxicity test with the marine polychaete embryos, HCl rather than phosphoric acid was used to adjust the mobile phase of HPLC to pH 2. This was because phosphate levels over 0.005 mol L^{-1} would adversely affect the development of the embryos, while chloride levels up to 0.85 mol L^{-1} would not adversely affect the marine polychaete embryos⁵⁵. Regarding the use of H₂SO₄ and HNO₃ for sample acidification prior to the analysis of UPLC/ESI-tqMS and total organic halogen, we followed the established protocols in Liu et al.²⁸ and the Standard Method 5320B⁵², respectively. We have added the information in the revised manuscript:

Lines 404–408 in the revision-marked manuscript, “A Waters UPLC system coupled with ESI-tqMS was used to characterize the degradation products of 2,6-DCQ. The degradation of 2,6-DCQ in seawater was simulated by adding 10 mg L^{-1} 2,6-DCQ to 50 mL ultrapure water, which was adjusted to pH 8.2 with 5 mM sodium carbonate (to simulate the alkalinity in seawater and also to be compatible with subsequent UPLC/ESI-tqMS analysis).”

Lines 431–434 in the revision-marked manuscript, “The mobile phase consisting of water (pH 2, adjusted with hydrochloric acid) and methanol (60/40, v/v) was applied at a flow rate of 1.0 mL

min⁻¹. Hydrochloric acid was used for water acidification to preclude the impact on the toxicity test with *P. dumerilii* embryos⁵⁵”.

Lines 442–445 in the revision-marked manuscript, “For this purpose, the aliquot was dissolved in 2 mL ultrapure water and pretreated following a previously reported method²⁸. Briefly, the aliquot was acidified to pH 2 with sulfuric acid (10%, v/v), added with sodium sulfate to saturation, and extracted with 2 mL methyl tert-butyl ether (MtBE)”.

Lines 454–456 in the revision-marked manuscript, “Briefly, the aliquot was dissolved in 100 mL ultrapure water and acidified to pH 2 with concentrated nitric acid⁵²”.

- “28. Liu, J., Zhang, X. & Li, Y. Photoconversion of chlorinated saline wastewater DBPs in receiving seawater is overall a detoxification process. *Environ. Sci. Technol.* **51**, 58–57 (2017).
52. APHA, AWWA & WEF. *Standard methods for the examination of water and wastewater*. Washington, DC, ed. 22nd (2012).
55. Han, J. & Zhang, X. Evaluating the comparative toxicity of DBP mixtures from different disinfection scenarios: a new approach by combining freeze-drying or rotoevaporation with a marine polychaete bioassay. *Environ. Sci. Technol.* **52**, 10552–10561 (2018).”

4. Authors have used sea water for studies. Is there any effect of different concentration of chloride ions on the activity or stability of 2,6 DCQ?

RESPONSE: A nice point. For the antimicrobial activity of 2,6-DCQ, the tests were conducted in phosphate buffered saline, which was used to maintain pH and prevent cells from rupturing due to osmosis. For the stability of 2,6-DCQ in aquatic environments, we used seawater because seawater is the immediate or ultimate receiving water body for municipal wastewater and urban runoff. To investigate the effect of chloride ions on the stability of 2,6-DCQ, we have conducted the degradation experiments of 2,6-DCQ in seawater (pH 8.2) and ultrapure water (added with 4 mM phosphate buffer at pH 8.2) in darkness. The results showed that the high chloride level in seawater had no discernible effect on the degradation of 2,6-DCQ (Supplementary Fig. 3). We have added the information in the revised manuscript.

Lines 185–188 in the revision-marked manuscript, “We also investigated the effect of the high chloride level in seawater on the degradation of 2,6-DCQ in darkness. The results showed that chloride ions had no discernible effect on the degradation of 2,6-DCQ (Supplementary Fig. 3).”

Supplementary Fig. 3 | Degradation of 2,6-DCQ at pH 8.2 in seawater and phosphate-buffered water in darkness.

5. Earlier articles on toxicity of 2,6 DCQ have reported that even few $\mu\text{g/L}$ concentration is toxic for human as well as aquatic lives (Environmental Pollution 245 (2019)719-724, Ecotoxicology and Environmental Safety 263 (2023) 115357). However, the authors have used up to 35 mg/L. How is its activity at lower concentrations?

RESPONSE: Thank you for the comment. As our results showed that by using 2,6-DCQ for disinfection, the survival curves of the tested pathogens followed the Chick-Watson law, if lower concentrations of 2,6-DCQ were used for the same contact time, lower inactivation of pathogens would be achieved.

Regarding the toxicity of 2,6-DCQ to humans, we would like to indicate that we also recognize the importance of understanding the potential human toxicity of 2,6-DCQ. The study (Ecotoxicol. Environ. Saf. 2023, 263, 115357) on the toxicity of 2,6-DCQ used an *in vitro* cell line model. Although such a study may be helpful for understanding the toxicity mechanism at the cellular level, it generally shows very limited relevance to the human response, and EPA has never regulated any DBPs based only on *in vitro* toxicological data. This is because “A cell is not a tissue, a cell is not an organ and a cell is not an organism. A chemical that is genotoxic or cytotoxic in cell culture does not necessarily mean it will induce adverse health effects in humans.” (Quoted from Richardson and Plewa. *J. Environ. Chem. Eng.* 2020, 8, 103939). Instead, mammalian animal toxicity testing is commonly used for understanding and comparing the toxic potency of different chemicals to humans. According to the Hodge and Sterner toxicity scale⁴⁸, a widely used scale for classifying chemicals based on the oral median lethal doses (LD_{50}) to rats, both 2,6-DCQ and PCMX are classified as “slightly toxic” to humans^{49,50}.

Regarding the toxicity of 2,6-DCQ to aquatic lives (Environ. Pollut. 2019, 245, 719), it is important to clarify that the toxicity of a chemical depends on both its inherent toxic potency and exposure concentrations. As indicated in the Introduction, PCMX has been detected in aquatic environments

at up to 9.57 $\mu\text{g L}^{-1}$. Considering that 2,6-DCQ is 9–22 times more efficient than PCMX in inactivating pathogens, the in-use concentration of 2,6-DCQ can be 9–22 times lower than PCMX; more importantly, we have shown that 2,6-DCQ can rapidly degrade in aquatic environments. These suggest that the exposure of 2,6-DCQ to aquatic lives should be much lower than PCMX when it is used as a disinfectant. In our study, we evaluated the developmental toxicity of 2,6-DCQ and PCMX to embryos of the marine polychaete *P. dumerilii* by combining their in-use concentrations and the toxic potency. The results showed that due to the rapid degradation of 2,6-DCQ in seawater, the toxicity of 2,6-DCQ (essentially a mixture of 2,6-DCQ and its degradation products) decreased dramatically, with its toxicity being insignificantly different from PCMX after 2 hours and 31 times lower than PCMX after 48 hours.

To echo your comment, we have added such information in the revised manuscript. We have also indicated in the revised manuscript that further studies on the human toxicity and environmental impact of 2,6-DCQ are necessary.

Lines 256–257 in the revision-marked manuscript, “Forty-eight hours after discharge into seawater, 2,6-DCQ exhibited 31 times lower toxicity to *P. dumerilii* embryos than PCMX”.

Lines 278–289 in the revision-marked manuscript, “It needs mentioning that although 2,6-DCQ exhibited stronger antimicrobial activities against pathogens than PCMX, according to the Hodge and Sterner toxicity scale (including extremely toxic, highly toxic, moderately toxic, and slightly toxic)⁴⁸, both chemicals are classified as substances “slightly toxic” to humans based on their median lethal doses to mammalian animals (LD_{50} ; oral, rat)^{49,50}. In addition, the much stronger antimicrobial activities of 2,6-DCQ than PCMX indicate much lower in-use concentrations of 2,6-DCQ when it is used as a disinfectant. Considering the comparable toxic potency of the two chemicals and the lower in-use concentrations of 2,6-DCQ, the toxicity potential of 2,6-DCQ to humans is probably lower than that of PCMX. Even so, the safety precautions that apply to PCMX or other disinfectants should also apply to 2,6-DCQ. Future studies may be conducted to systematically evaluate the human toxicity and environmental impact of 2,6-DCQ.”

“48. Hodge, H. C. & Sterner, J. H. Tabulation of toxicity classes. *Am. Ind. Hyg. Assoc.* **10**, 93–96 (1949).

49. National Research Council. The Chemical-Biological Coordination Center of the National Research Council (The National Academies Press, 1953).

50. Liebert, M. A. Final report on the safety assessment of chloroxylenol. *J. Am. Coll. Toxicol.* **4**, 147–169 (1985).”

6. Is 2,6 DCQ cost effective in comparison to commercial PCMX?

RESPONSE: A great point. As 2,6-DCQ has not been mass produced for commercial application, we propose to evaluate the cost of 2,6-DCQ in comparison with PCMX by considering the materials and energy required in the synthesizing processes^{46,47}. In brief, to obtain 1 kg of PCMX, 2.63 kg of 3,5-dimethylphenol is placed in a suitable vessel with a stirrer, and 1.31 kg of sulfuryl

chloride (SO₂Cl₂) is slowly added. The temperature is kept at 40 °C during the reaction. When all the SO₂Cl₂ has been added, the reaction mixture is heated to 80 °C and the volatile impurities in it are removed by air blowing. Part of the PCMX is separated from the mother liquor during cooling, and further PCMX can be isolated by vacuum distillation of the mother liquor and further crystallization. A total of 1 kg of PCMX can be obtained. To obtain 1 kg of 2,6-DCQ, 1.18 kg of 2,4,6-trichlorophenol is dissolved in 3.59 kg of methanol. The dissolved 2,4,6-trichlorophenol is added dropwise to 1200 mL of 5% (w/w) nitric acid in a closed reaction vessel with a stirrer. The reaction is carried out at 20 °C with an oxygen supply of 2.0 atm (i.e., 0.013 kg O₂ for a 5 L reaction vessel). After a reaction time of 2 hours, 1 kg of 2,6-DCQ can be obtained by filtration of the mother liquor. The unit prices of the raw materials are from a global supplier Alfa Aesar (note that unit prices can be significantly lower for larger quantities). According to the synthesis flowcharts shown below, the raw material cost for synthesizing 2,6-DCQ is lower than that for synthesizing PCMX. Since no heating or cooling processes are required for 2,6-DCQ synthesis, the energy consumption should also be lower than that for PCMX. Accordingly, 2,6-DCQ should be cost effective compared to PCMX. We have supplemented the cost analysis in the revised manuscript.

Lines 270–273 in the revision-marked manuscript, “We also conducted comparative cost analysis for 2,6-DCQ and PCMX. Based on the consumption of raw materials and energy in the synthesis processes^{46,47}, the cost for producing 2,6-DCQ should be lower than that for producing PCMX (Supplementary Fig. 9).”

Supplementary Fig. 9 | Synthesis flowcharts of (a) PCMX and (b) 2,6-DCQ. The unit prices of raw materials were obtained from a global supplier Alfa Aesar (<https://alfaesar.com/>).

- “46. Gladden, G. W. Preparation of 2-chlor-meta-5-xyleneol. US Patent No. 2,350,677 (1944).
 47. Xu, Y. Preparation method of 1,4-cyclohexanedione. CN Patent No. 109942388A (2019).”

REVIEWERS' COMMENTS

Reviewer #1 (Remarks to the Author):

The reviewers' comments/suggestions were all well addressed in the revised manuscript. The authors' efforts are appreciated. I do not have further comments, and believe it is ready for publication.

Reviewer #2 (Remarks to the Author):

It can be accepted as all the comments have been implemented